# Dual-Template Magnetic Molecularly Imprinted Polymer for Simultaneous Determination of Spot Urine Metanephrines and 3-Methoxytyramine for the Diagnosis of Pheochromocytomas and Paragangliomas

**DOI:** 10.3390/molecules27113520

**Published:** 2022-05-30

**Authors:** Hongyu Zeng, Xiaoqing Zhang, Qianna Zhen, Yifan He, Haoran Wang, Yang Zhu, Qi Sun, Min Ding

**Affiliations:** 1Key Laboratory of Laboratory Medical Diagnostics, Ministry of Education, School of Laboratory Medicine, Chongqing Medical University, Chongqing 400016, China; 2019110854@stu.cqmu.edu.cn (H.Z.); xzhang5@cqmu.edu.cn (X.Z.); marinew329@gmail.com (H.W.); 2019110870@stu.cqmu.edu.cn (Y.Z.); 2021110558@stu.cqmu.edu.cn (Q.S.); 2Department of Endocrinology, The First Affiliated Hospital of Chongqing Medical University, Chongqing 400016, China; zqn19831021@163.com (Q.Z.); mors_hyf@hotmail.com (Y.H.)

**Keywords:** magnetic molecularly imprinted polymer, normetanephrine, metanephrine, 3-methoxytyramine, spot urine, PPGLs diagnosis

## Abstract

A novel dual-template magnetic molecularly imprinted polymer (MMIP) was synthesized to extract normetanephrine (NMN), metanephrine (MN) and 3-methoxytyramine (3-MT) from spot urine samples. As the adsorbent of dispersive solid-phase extraction (d-SPE), the MMIP was prepared using dopamine and MN as dual templates, methacrylic acid as the functional monomer, ethylene glycol dimethacrylate as the crosslinking reagent and magnetic nanoparticles as the magnetic core. NMN, MN, 3-MT and creatinine (Cr) in spot urine samples were selectively enriched by d-SPE and detected by HPLC-fluorescence detection/ultraviolet detection. The peak area (A) ratios of NMN, MN and 3-MT to Cr were used for the diagnosis of pheochromocytomas and paragangliomas (PPGLs). The results showed that the adsorption efficiencies of MMIP for target analytes were all higher than 89.0%, and the coefficient variation precisions of intra-assay and inter-assay for the analytes were within 4.9% and 6.3%, respectively. The recoveries of the analytes were from 93.2% to 112.8%. The MMIP was still functional within 14 days and could be reused at least seven times. The d-SPE and recommended solid-phase extraction (SPE) were both used to pretreat spot urine samples from 18 PPGLs patients and 22 healthy controls. The correlation coefficients of A_NMN_/A_Cr_ and A_MN_/A_Cr_ between d-SPE and SPE were both higher than 0.95. In addition, the areas under the receiver operator curves for spot urine A_NMN_/A_Cr_, A_MN_/A_Cr_ and plasma free NMN and MN were 0.975, 0.773 and 0.990, 0.821, respectively, indicating the two methods had the similar performances. The d-SPE method took only 20 min, which was effective in clinical application.

## 1. Introduction

Pheochromocytomas and paragangliomas (PPGLs) are neoplasms arising from adrenal and extra-adrenal chromaffin tissues, respectively [1]. Typically, PPGLs usually secrete excessive catecholamines including dopamine (DA), epinephrine (E) and norepinephrine (NE). Overproduced catecholamines could seriously damage patients’ cardiovascular system, which leads to symptoms such as hypertension, headache and sweating. Some PPGLs patients even develop hypertensive crises, which are the primary cause of death for PPGLs patients [2,3]. Most PPGLs are nonmalignant and can be cured by operative treatment [4]. Therefore, a fast and accurate diagnosis of PPGLs is important.

Normetanephrine (NMN) and metanephrine (MN) are O-methylated metabolites of NE and E, respectively. Plasma free and 24 h-fractionated metanephrines (MNs) are recommended by the Endocrine society for patients suspected of PPGLs [5]. In addition, elevated NMN is concerned with neuroblastoma [6,7,8]. 3-methoxytyramine (3-MT) is the O-methylated metabolite of DA and is conventionally determined for the differential diagnosis of dopamine hypersecretory tumors. Liu’s group [7] found that MNs and 3-MT used as a combined biomarker could increase the diagnostic sensitivity of PPGLs, and several groups have reported that 3-MT was useful for the diagnosis of neuroblastoma [9,10].

Plasma and 24 h-fractionated urine MNs detected by HPLC-mass spectrometry/electrochemical detection (ECD) were recommended in clinic. However, blood sampling requires patients to lie still for at least 30 min, and 24 h urine samples were inconvenient to collect for outpatients. To solve this problem, our previous work established a method using HPLC-fluorescence detection/ultraviolet detection (FLD/UVD) to determine chromatographic peak area (A) ratios of MNs and 3-MT to Cr (creatinine) (A_MNs_/A_Cr_ and A_3-MT_/A_Cr_) in spot urine samples for the accurate diagnosis of PPGLs. The collection of spot urine samples is fast and convenient. Moreover, the diagnostic efficiencies of spot urine A_MNs_/A_Cr_ were higher than those of plasma MN levels, and spot urine A_3-MT_/A_Cr_ was valuable in diagnosing malignant PPGLs [11].

The present sample pretreatment method for detecting plasma free MNs and 24 h-fractionated MNs is solid-phase extraction (SPE) [12,13,14,15,16]. However, SPE is tedious and time consuming. As a result, the turnaround time of this test was too long, which made patients spend more on treatment costs and time [17,18].

Molecular imprinted polymer (MIP) is a three-dimensional polymer containing recognition cavities complementary to templates in size, shape and structure, which could mimic the antigen–antibody reaction [19,20,21,22], and it has been proven to offer excellent selectivity for compounds with similar structures as the template used in synthesis, which makes it appropriate for complex sample pretreatments [23,24,25,26,27,28,29,30,31,32].

In recent years, dispersive solid-phase extraction (d-SPE) has been one of the most popular and applicable techniques in sample preparation because it can easily separate analytes from aqueous mediums such as urine samples [33,34,35,36,37,38,39]. Magnetic molecularly imprinted polymer (MMIP) is a compound prepared by encapsulating inorganic magnetic nanoparticles with polymers. MMIP possesses the merits of magnetic nanoparticles and MIP, which is an ideal adsorbent for d-SPE [20,21,32]. Application of d-SPE based on MMIP can extract specific molecules selectively and effectively from diverse matrices. Compared with SPE, d-SPE can shorten the time of pretreatment and simplify the pretreatment process significantly. Moreover, MMIP is a stable material and can be regenerated several times [40,41,42,43]. Therefore, d-SPE is simple, fast and cost-effective compared with SPE.

Herein, a novel kind of MMIP based on magnetic nanoparticles using DA and MN as dual templates, methacrylic acid (MAA) as functional monomers, ethylene glycol dimethacrylate (EGDMA) as the crosslinking reagent and azobisisobutyronitrile (AIBN) as the initiator was prepared. Furthermore, it was used as sorbent in the d-SPE process to pretreat spot urine samples. Then, HPLC-FLD/UVD was used to detect spot urine A_MNs_, A_3-MT_ and A_Cr_. The spot urine A_MNs_/A_Cr_ had similar diagnostic efficiencies for the diagnosis of PPGLs as plasma free MNs.

## 2. Materials and Methods

### 2.1. Chemicals and Material

Magnetic nanoparticles (20–30 nm) were purchased from Jinpan biotech (Shanghai, China). HPLC-grade methanol and acetonitrile were purchased from Tedia (Fairfield, OH, USA).

3-methacryloxypropyltrimethoxysilane (MPS), AIBN, MAA and EGDMA were purchased from Aladdin (Shanghai, China).

Creatinine (Cr), dopamine hydrochloride (DA·HCl), DL-E·HCl, DL-NE·HCl, 3-MT·HCl and DL-NMN·HCl were purchased from Sigma-Aldrich (Saint Louis, USA). DL-MN·HCl was purchased from Mucklin (Shanghai, China).

Ethanol, acetic acid, sodium dihydrogen phosphate, disodium hydrogen phosphate, trisodium phosphate and acetic acid were purchased from Chuandong chemical (Chongqing, China), and water was purified by a Milli-Q system from Merck (Darmstadt, Germany). All other chemicals were of analytical grade.

### 2.2. Patients

Spot urine samples of 18 PPGLs patients and 22 healthy controls were collected from July 2018 to February 2020 in the First Affiliated Hospital of Chongqing Medical University. Patients were asked not to take anti-depression and acetaminophen medicine for 5 days before the sample collection. In addition, the intake of apples, bananas, tea and coffee was forbidden for 12 h to guarantee the accuracy of the results. All of the spot urine samples were stored at −20 °C before use.

The study was approved by the ethical committee of the First Affiliated Hospital of Chongqing Medical University. The concentrations of patients’ plasma free MNs were detected by HPLC—ECD [15] before the urine collection. Meanwhile, computed tomography was used in the auxiliary diagnosis of PPGLs. The definite diagnosis of PPGLs was performed by postoperative pathological examinations.

### 2.3. Synthesis of MMIP and Magnetic Non-Imprinted Polymer

First, 50.0 mg magnetic nanoparticles, 2.0 mL of water and 2.0 mL of ethanol were successively added into a 50 mL centrifuge tube, respectively. The tube was shaken violently until the particles were evenly distributed in the solution, then 4.0 mL of MPS was dropwise added into the tube. In a nitrogen atmosphere, the tube was placed into a thermostatic water bath at 40 °C shaking for 12 h to modify magnetic nanoparticles with a double bond. After the reaction, the modified magnetic nanoparticles were cleaned with ethanol 3 times and then placed in an incubator at 37 °C to dry.

After dehydration overnight, 50.0 mg Fe_3_O_4_@SiO_2_-CH=CH_2_ nanoparticles were accurately weighed and added with 9.5 mL of acetonitrile and 0.5 mL of water, respectively. Then, 0.015 mmol DA and 0.010 mmol MN as templates, 1.000 mmol MAA as functional monomer, 1.000 mmol EGDMA as crosslinking reagent and 13.0 mg AIBN as initiator were successively added to 50 mL tubes. In a nitrogen atmosphere, the tubes were put into a thermostatic water bath at 65 °C shaking for 24 h. After the synthesis, MMIP was eluted with acetic acid (2.0%, *v*/*v*) 3 times for template removal and placed in the incubator at 37 °C before use. The synthesis of magnetic non-imprinted polymer (MNIP) was carried out without templates and the rest protocols were the same as MMIP.

### 2.4. Performance of d-SPE

First, 500 μL of spot urine sample and 40 mg of MMIP were added to 2.5 mL of PBS (0.1 mmol/L, pH 9.0), respectively. After shaking violently for 30 s and incubating for 8 min, MMIP was separated from the matrix, and the supernatant was discarded. Finally, 200 μL of acetic acid (2%, *v*/*v*) was added. After shaking violently for 30 s and incubating for 7.5 min, MMIP was separated from the matrix, and 50 μL of the supernatant was detected by HPLC-FLD/UVD. MMIP was rinsed with water and dried for regeneration.

### 2.5. Chromatographic Conditions

The chromatographic experiment was performed on Agilent 1100 HPLC equipped with a G1321A fluorescence detector and a G1314A ultraviolet detector (Palo Alto, CA, USA). The chromatographic conditions were established based on our previous work with a slight modification [11]. Separation of analytes was achieved at 30 °C with a Shim-Pack VP-ODS column (150 mm × 4.6 mm i.d., 4.6 µm, Shimadzu, Kyoto, Japan), which was attached to a C18 security guard column (4 mm × 3 mm i.d., 5 µm, Phenomenex, California, USA). The mobile phase consisted of 70 mmol/L sodium dihydrogen phosphate (solvent A) and methanol (solvent B). The mobile phase was set at 0.8 mL/min with the gradients as follows: The content of B increased from 5% to 10% within 4.0 min and maintained for 8.0 min. The excitation and emission wavelengths of fluorescence detection were set at 278 nm and 320 nm, respectively. The wavelength of ultraviolet detection was set at 235 nm.

## 3. Results and Discussion

### 3.1. MMIP Synthesis

The synthesis of MMIP consisted of three steps including modification, thermal radical polymerization and template removal, as shown in Figure 1. The MPS was used as a modifier to induce double bonds on magnetic nanoparticles, and then MIP was coated with nanoparticles by thermal radical polymerization. After the reaction, templates were removed by breaking the hydrogen bonds using acetic acid to reduce the pH value. After template removal, specific three-dimensional caves complementary to the templates were exposed, which could specifically adsorb analytes.

### 3.2. Optimization of MMIP Synthesis

E, NE, DA, MN, NMN and 3-MT were selected as templates in the synthesis of MMIP, and the results showed that when using a single kind of template, MMIP using DA as the template had the best adsorption efficiencies (*E_A_*), as shown in Table 1. This is because DA contains the structure of catechol, which is capable of forming hydrogen bonds. As a result, the MMIP using DA as the template had the highest *E_A_* when extracting analytes in standard solution. However, when used to extract spot urine samples, the *E_A_* of MN by MMIP using MN as the template was far better than that of MMIP using DA as the template. The reason may be that the MMIP using DA was low in the selectivity to MN, and the matrix in spot urine samples competitively combined the adsorption sites on MMIP. As a result, DA and MN were chosen as dual templates to synthesize MMIP in order to increase the *E_A_* of MN in spot urine samples. The results of MMIP and MNIP adsorbing spot urine samples were shown in Table 2. Different ratios of templates to functional monomers were explored, and the highest *E_A_* was obtained when the ratio was selected at 1:20, as shown in Appendix A. The ideal ratio of functional monomers to crosslinking reagent was selected at 1:1, as shown in Appendix A.

*E_A_* was calculated based on Formula 1
*E_A_* = (*x*_1_ − *x*_2_)/*x*_1_ × 100%(1)

*x*_1_ means the peak area of analytes standards before the adsorption.

*x*_2_ means the peak area of analytes standards in the supernatant after the adsorption.

### 3.3. The Characteristics of MMIP

Fourier Transform Infrared Spectroscopy (FTIR) was conducted on SU8020 (Hitachi, Tokyo, Japan) and a scanning electron microscope (SEM) was performed on IS10 (Nicolet, Thermo Fisher Scientific, Waltham, MA, USA), respectively. As shown in Figure 2, the MMIP and magnetic nanoparticles were scanned by FTIR between the wavelengths of 450 to 4000 nm; the absorption peaks at 560.64 nm, 1011.59 nm, 1628.29 nm and 3391.87 nm are the vibrations of Fe-O, C-O, C=O and –OH, respectively. Fe-O exists in magnetic nanoparticles, C-O and C=O exist in MPS, MAA and EGDMA, while -OH exists in water and MAA. The FTIR results showed that the MMIP was successfully synthesized based on magnetic nanoparticles. The SEM picture showed that the MIP with a porous structure was evenly coated with nanoparticles, which made it conducive to the specific adsorption of target analytes.

### 3.4. The Optimization of d-SPE Conditions

The specific recognition between analytes and MMIP is based on hydrogen bonds, which are highly affected by pH value. To evaluate the adsorption efficiencies of MMIP to analytes at different pH values, a series of 0.01 mol/L PBS with different pH values were prepared for d-SPE. When pH ≥ 8.0, the *E_A_* of NMN, MN and 3-MT were stable and satisfactory. As the pK_a_ of NMN, MN and 3-MT are 9.06, 9.25 and 9.64, respectively, the analytes will ionize, which makes it difficult to form hydrogen bonds with MMIP at pH < 8.0. However, at pH ≥ 8.0, the analytes mainly exist as molecular forms, which makes it easier to form hydrogen bonds with MMIP and, therefore, improves the *E_A_* of analytes. As MMIP is unstable in strong alkaline environments, pH 9.0 was chosen as the optimal condition for the adsorption. Different adsorption times of d-SPE were explored. The *E_A_* of MMIP increased over time before 8.0 min. When MMIP was saturated for over 8.0 min, the *E_A_* was stabilized and time-independent. Therefore, 8.0 min was chosen as the optimal adsorption time. Acetic acid aqueous was used to regulate the pH value of eluant in d-SPE in order to break the hydrogen bonds between MMIP and the analytes. Different concentrations of acetic acid aqueous solution were used to determine the elution efficiency (*E_B_*) of MMIP, and *E_B_* was calculated according to Formula 2. When the concentration of acetic acid aqueous was ≥2.0%, the elution efficiency became stable at a high level. Because of the instability of MMIP in strongly acidic environments, the optimal concentration of acetic acid aqueous was selected at 2.0%. Different elution times were performed in the d-SPE process. When the elution time was less than 7.5 min, the *E_B_* of MMIP increased with time. When the *E_B_* of MMIP became saturated at 7.5 min, it remained stable. Therefore, the optimal time was selected at 7.5 min. The results of the optimization are shown in Appendix A.
*E_B_* = *x*_4_/*x*_3_ × 100%(2)

*x*_3_ means the peak area of analytes standards adsorbed on MMIP (*x*_1_ − *x*_2_).

*x*_4_ means the peak area of analytes standards in eluant.

### 3.5. Precision, Usage Count and Stability of MMIP

The pooled spot urine samples of PPGLs patients and healthy controls were used to test the precision of MMIP five times, respectively. The coefficients of variations (CV) of MNs and 3-MT in spot urine samples from healthy controls and PPGLs patients using MMIP from the same batch were all less than 4.9%. When using MMIP from different batches, the CVs of MNs and 3-MT were all less than 6.3%, as shown in Appendix A. To test the reuse times of the MMIP, the pooled spot urine samples from healthy controls and PPGLs patients were divided into seven parts, respectively, and the MMIP was used to extract these samples in a blind order. As shown in Appendix A, the CVs of PPGLs patients and the healthy control were lower than 4.6% and 5.5%, respectively, which meant that the MMIP could be used at least 14 times. In addition, the results showed no cross contamination between the PPGLs group and the control group. Then, to test its long-term stability, MMIP was placed in a dry area for 1, 14 and 28 days, respectively. The MMIP stored for different times was used in the d-SPE process to extract spot urine samples three times, and as shown in Appendix A, the adsorption amount of MMIP reduced by 0.7–5.2% and 11.6–17.6% in 14 days and 28 days compared with MMIP stored for 1 day, respectively, which illustrated that MMIP was stable for at least 14 days. The chromatograms of spot urine samples from a patient with PPGLs and a healthy volunteer are shown in Figure 3.

### 3.6. Recoveries and Calibration Curves

Calibration samples were prepared by adding 20 μL of standard solution at a series of concentrations (25–1000 μg) to 480 μL of the pooled spot urine, respectively. The value *Y* in the calibration curve was obtained by the subtraction of the peak areas of the analytes after the calibration addition to the peak areas of the analytes before the calibration addition; the value *X* in the calibration curve represented the concentration of the analytes added to the samples. As shown in Appendix A, the peak area of MNs and 3-MT showed a linear relation with its concentration in the range of 25 to 1000 μg/L, *R*^2^ > 0.999. The limits of detection of NMN, MN and 3-MT were 4.2 μg/L, 9.6 μg/L and 7.7 μg/L, respectively, and the limits of quantification were 15.3 μg/L for NMN, 24.2 μg/L for MN and 21.3 μg/L for 3-MT. To test the recoveries of the analytes, the standard solution of NMN, MN and 3-MT at high (800 μg/L), medium (250 μg/L) and low (50 μg/L) levels were added to the pooled spot urine samples, respectively, and each concentration was performed three times. As shown in Appendix A, the recoveries of the analytes in the spot urine samples were in the range of 93.2–112.8%, which indicates satisfactory recoveries.

### 3.7. The Methods Comparison between SPE and d-SPE

The developed method was applied for analyzing the spot urine samples from 18 PPGLs patients and 22 healthy controls. To compare the A_MNs_/A_Cr_ and A_3-MT_/A_Cr_ results of the spot urine samples pretreated by d-SPE and SPE [11], Spearman’s correlation test was performed. The coefficients of A_MNs_/A_Cr_ were both higher than 0.95 (*r* = 0.961 and 0.993), as shown in Figure 4, indicating that the A_MNs_/A_Cr_ results by two methods were highly related; the A_3-MT_/A_Cr_ results of the two methods were moderately related, showing no significant difference between the PPGLs group and the control group (*p* > 0.05). The results are shown in Appendix A. Compared with SPE, d-SPE takes only 20 min, which largely shortens the sample turnaround time and increases the efficiencies of the pretreatment. Moreover, MMIP can be reused at least 14 times. SPE columns are disposable consumables, which means d-SPE is more cost-effective.

### 3.8. The Diagnosis of PPGLs

To evaluate the diagnostic efficiency of the established method, the spot urine A_MNs_/A_Cr_ and plasma free MNs from 18 PPGLs and 22 healthy controls were determined by the established method and the recommended SPE-HPLC-ECD [1] method, respectively, as shown in Figure 5; the data were used to draw the receiver operator characteristic curve, as shown in Figure 6. The diagnostic efficiencies were obtained by the areas under the operator characteristic curves (AUC). The diagnostic efficiencies of spot urine A_NMN_/A_Cr_ and A_MN_/A_Cr_ were 0.975 and 0.773, respectively, and the diagnostic efficiencies of plasma free NMN and MN were 0.990 and 0.821, respectively, which shows that the two methods had the similar performances. The sensitivity and specificity of plasma MNs and the spot urine free A_MNs_/A_Cr_ for the diagnosis of PPGLs are shown in Appendix A, indicating that the spot urine A_MNs_/A_Cr_ determined by d-SPE-HPLC-FLD/UVD was effective for the diagnosis of PPGLs.

## 4. Conclusions

In this study, MMIP was prepared by the thermal radical polymerization using DA and MN as dual templates, MAA as the functional monomers, EGDMA as the crosslinking reagent, AIBN as an initiator and nanoparticles as carriers. The synthesis of MMIP is simple and cost effective. The d-SPE process based on MMIP can realize simultaneous enrichment of MNs, 3-MT and Cr in spot urine samples. Furthermore, A_MNs_/A_Cr_ determined by d-SPE-HPLC-FLD/UVD had similar diagnostic efficiencies for the diagnosis of PPGLs as plasma free MNs determined by SPE-HPLC-ECD. The d-SPE process took only 20 min, superior to the traditional SPE, and therefore has a great potential in clinical application.

## Figures and Tables

**Figure 1 molecules-27-03520-f001:**
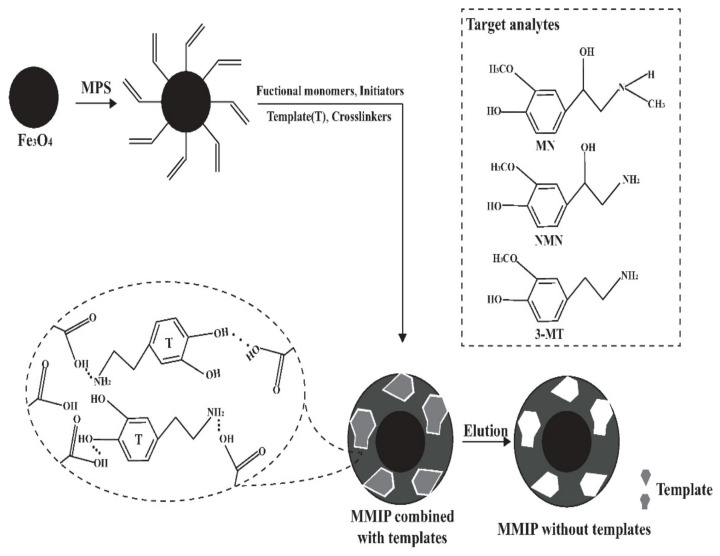
Schematic diagram of MMIP synthesis.

**Figure 2 molecules-27-03520-f002:**
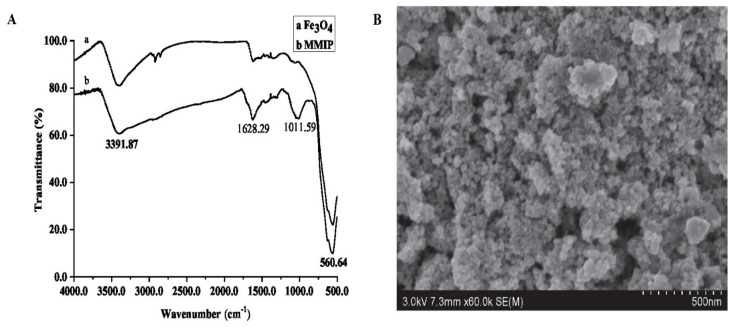
The characteristics of MMIP ((**A**). FTIR; (**B**). SEM).

**Figure 3 molecules-27-03520-f003:**
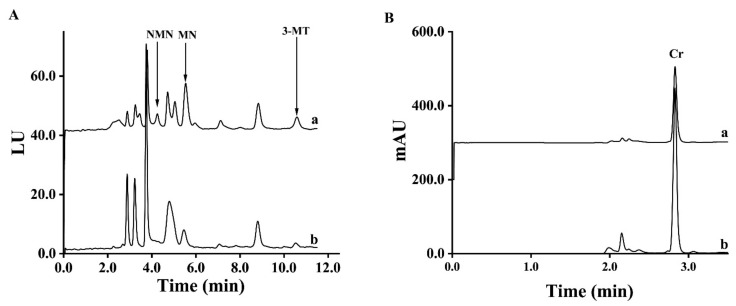
Chromatograms of spot urine samples from a PPGLs patient (a) and a healthy control (b) ((**A**). FLD; (**B**). UVD).

**Figure 4 molecules-27-03520-f004:**
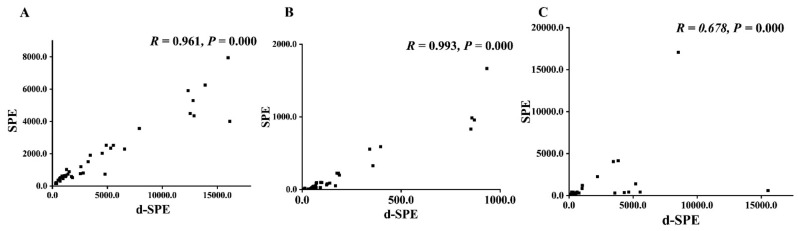
The comparison of SPE and d-SPE ((**A**). A_NMN/_A_Cr_; (**B**). A_MN/_A_Cr_; (**C**). A_3-MT/_A_Cr_).

**Figure 5 molecules-27-03520-f005:**
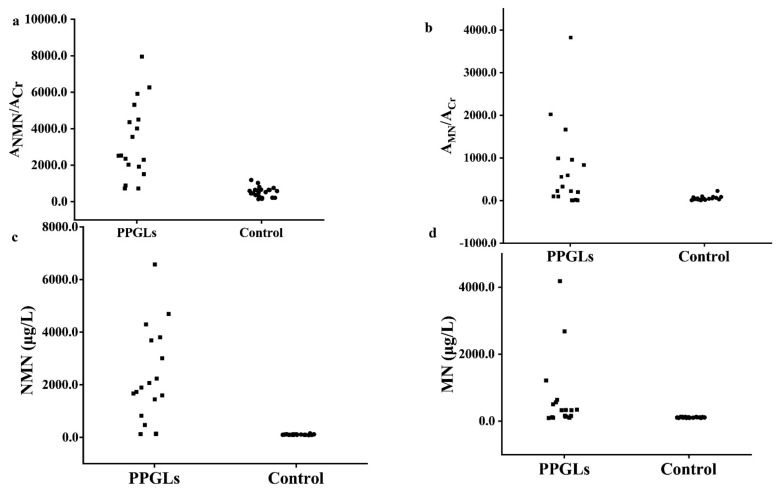
The spot urine A_MNs_/A_Cr_ results and plasma free MNs results of PPGLs and control ((**a**). spot urine A_NMN/_A_Cr_; (**b**). spot urine A_MN/_A_Cr_ (**c**). plasma free NMN (**d**). plasma free MN).

**Figure 6 molecules-27-03520-f006:**
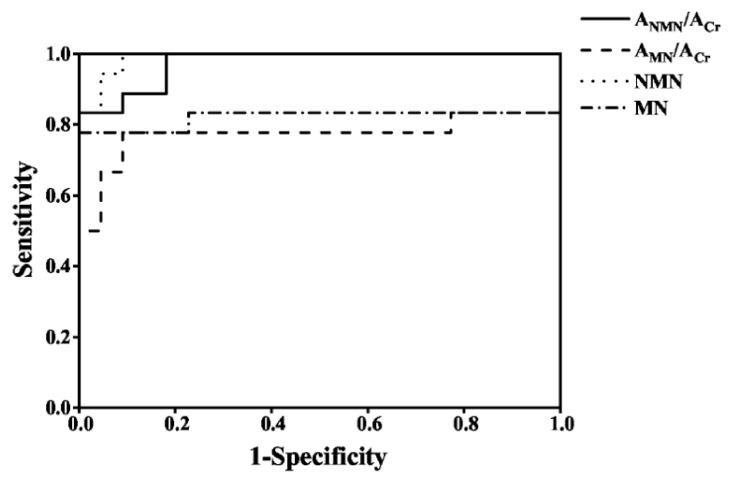
The operator characteristic curves of spot urine A_MNs_/A_Cr_ and plasma free MNs for the diagnosis of PPGLs.

**Table 1 molecules-27-03520-t001:** The adsorption efficiency of synthesized MMIP of different template molecules. ($\bar{x}$ ± *s*, *n* = 3).

Template	*E_A_* (%) of Standards		
	NMN	MN	3-MT
DA	94.5 ± 0.0	90.8 ± 0.7	94.9 ± 0.1
E	78.1 ± 1.3	73.4 ± 0.6	82.0 ± 0.6
NE	84.6 ± 0.3	75.8 ± 0.9	87.3 ± 0.6
NMN	52.5 ± 2.2	46.6 ± 0.3	61.5 ± 2.3
MN	57.8 ± 1.1	64.3 ± 1.2	48.0 ± 0.6
3-MT	56.4 ± 0.9	32.7 ± 1.4	45.7 ± 2.2
DA +MN	91.3 ± 0.1	89.4 ± 0.2	94.6 ± 0.0
MNIP	8.9 ± 1.2	9.4 ± 0.5	8.5 ± 0.5

**Table 2 molecules-27-03520-t002:** Enrichment results of spot urine samples by MMIP synthesized with different template molecules. ($\bar{x}\;\pm$
*s*, *n* = 3).

Template	Peak Area		
	NMN	MN	3-MT
DA	287.5 ± 8.3	2.1 ± 0.1	44.1 ± 1.6
E	12.2 ± 0.5	12.0 ± 0.4	10.1 ± 0.1
NE	3.8 ± 0.3	3.6 ± 0.3	82.9 ± 7.3
NMN	26.3 ± 0.5	0.0	5.0 ± 0.5
MN	5.6 ± 0.2	42.9 ± 0.3	49.0 ± 0.8
3-MT	24.6 ± 0.1	29.3 ± 0.8	51.1 ± 0.7
DA + MN	25.8 ± 1.0	40.3 ± 1.2	37.4 ± 1.8
MNIP	3.2 ± 0.1	5.6 ± 0.1	3.0 ± 1.6

## Data Availability

The data presented in this study are available on request from the corresponding author. The data are not publicly available due to it was protected by patent ‘CN 2017107169315′.

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
