# Peer review of "Dual-Template Magnetic Molecularly Imprinted Polymer for Simultaneous Determination of Spot Urine Metanephrines and 3-Methoxytyramine for the Diagnosis of Pheochromocytomas and Paragangliomas"

_molecules, 2022, doi:10.3390/molecules27113520_

Round 1
Reviewer 1 Report
The article “Zeng H et al.: Dual-template Magnetic Molecularly Imprinted Polymer for Simultaneous Determination of Spot Urine Metanephrines and 3-methoxytyramine for the Diagnosis of Pheochromocytomas and Paragangliomas.“ is interesting, the authors describe the synthesis and use of dual-template (dopamine, metanephrine) magnetic molecularly imprinted polymer, which is used for dispersive solid phase extraction of metanephrines and 3-methoxytyramine from urine and their determination by liquid chromatography. The authors state that the procedure is clinically effective and its effectiveness corresponds to the determination of free metanephrines in plasma.
I have several reservations about the article:
1) A large number of similar abbreviations are used in the article, which can make it difficult to read. I recommend that abbreviations be written based on the frequency of occurrence of the terms. I would replace the less frequent with full names.
2) When using an abbreviation for the first time, the abbreviation in parentheses should follow the full wording of the term.
Line 22 - not specified as FLD/UVD,
Lines 29.30 - what are the coefficients ANMN / ACr, AMN/ACr,
Line 31 - what do the numbers in parentheses mean for plasma free NMN (0.990) and MN (0.821)?
3) Lines 61, 88 - As substantiated by the fact that "the diagnostic efficiency of spot urine AMNs / ACr were higher than those of plasma MNs levels". Is there any literature that sheds light on this?
4) Line 95 – repair 3-methacryloxypropyltrimethoxysi-lane
5) lines 98-101: The abbreviations DA, 3-MT, NMN and MN refer to the compounds but not their hydrochlorides. In the text it would be necessary to write eg DA.HCl.
6) line 129: The text says "0.025 mmol/l templates". It is necessary to specify the number and type of templates.
7) line 171: It is not explained what EA is.
8) lines 183, 184: I miss the explanation of "the peak area of standards and supernatant after adsorption".
9) line 189: In what units was the peak area determined?
10) lines 227, 228: Dtto lines 183, 184
11) line 231: From which number of repetitions the coefficients of variation were determined.
12) Fig. 3A: What does the designation LU on the Y axis mean? Is the 3-MT level the same in both the PPGL patient and the control?
13) line 258: You present LOD of NMN, MN, 3-MT. Has a limit of quantification also been set?
14) lines 284, 285: How was the diagnostic efficiencies of the spot urine AMNs / ACr and plasma free MNs determined? Does the first number correspond to MN, the second number to NMN, what do the numbers mean? Why are the spot urine MN, NMN, 3-MT related to creatinine?
15) line 291: Repair plasama
16) Fig. 6: From how many patients were ROC curves generated? The image is not clear.
17) lines 299, 300: A comparison of the diagnostic efficiency of AMNs/ACr determined by d-SPE-HPLC-FLD/UVD and plasma free MNs determined by SPE-HPLC-ECD would, in my opinion, require more than just 18 PPGL patients.
Reviewer 2 Report
This paper addresses a topical issue, namely the synthesis of magnetic molecularly imprinted polymer that can be used for the diagnosis of pheochromocytomas and paragangliomas. The strong points are the clear and concise description of the method of obtaining this product but also the test protocol on samples from several patients. A small disadvantage, in my opinion, is that the paper is written using ultra-specialized language that can be difficult to read for researchers in related fields. For example, the introduction could include more details about the condition to which this new method is addressed (L37-38).
Specific comments: in figures 4 and 5 the text is too small, please enlarge the font so that it can be read. In general, the abbreviations used in the tables must be described (for example in Table 1 DA, E, etc.). The same goes for the supplemental material. Also for the supplemental material to use the same number of decimals in the tables and to increase the font in the figure.
